# Swelling-strengthening hydrogels by embedding with deformable nanobarriers

Feng Wu[1], Yan Pang [2✉] & Jinyao Liu [1✉]

Biological tissues, such as muscle, can increase their mechanical strength after swelling due to the existence of many biological membrane barriers that can regulate the transmembrane transport of water molecules and ions. Oppositely, typical synthetic materials show a swelling-weakening behavior, which always suffers from a sharp decline in mechanical strength after swelling, because of the dilution of the network. Here, we describe a swelling-strengthening phenomenon of polymer materials achieved by a bioinspired strategy. Liposomal membrane nanobarriers are covalently embedded in a crosslinked network to regulate transmembrane transport. After swelling, the stretched network deforms the liposomes and subsequently initiates the transmembrane diffusion of the encapsulated molecules that can trigger the formation of a new network from the preloaded precursor. Thanks to the tough nature of the double-network structure, the swelling-strengthening phenomenon is achieved to polymer hydrogels successfully. Swelling-triggered self-strengthening enables the development of various dynamic materials.

[1] Shanghai Key Laboratory for Nucleic Acid Chemistry and Nanomedicine, Institute of Molecular Medicine, State Key Laboratory of Oncogenes and Related Genes, Shanghai Cancer Institute, Renji Hospital, School of Medicine, Shanghai Jiao Tong University, 200127 Shanghai, China. [2] Department of Ophthalmology, Shanghai Ninth People's Hospital, School of Medicine, Shanghai Jiao Tong University, 200011 Shanghai, China. ✉email: yanpang@shsmu.edu.cn; jyliu@sjtu.edu.cn

Biological tissues can increase their mechanical strength when needed[1–3], although they are essentially hydrogel materials and are immersed in body fluids[4,5]. For example, muscles become much stronger after swelling with blood[6,7]. The contraction of skeletal muscle can activate its associated muscle by increasing blood flow. The specific release of potassium and calcium ions from the activated muscle is able to regulate contractile tension, which can further enhance the hyperemia. Consequently, the enhanced hyperemia results in a large increase in muscle stiffness (Fig. 1a). Central to the swelling-strengthening nature of these systems is the existence of many biological membrane barriers that can regulate the transmembrane transport of water molecules and ions[8–10]. Namely, selective transport across these barriers during swelling maintains a steady structure of the network, which mainly determines the mechanical strength of the system[11–14]. By contrast, synthetic materials present a typical swelling-weakening phenomenon because of the dilution of the network[15–18], which always suffers from a sharp decline in mechanical strength after swelling and largely limits the application, particularly when a given mechanical strength is required, such as biological glues or artificial tissues[19–24]. Although studies have shown that a few specially designed networks with a hydrophilic–lipophilic balance can resist swelling[25–28], the preparation of polymer materials capable of reinforcing their mechanical strength after swelling remains difficult. The key challenge is that during swelling, the major change that occurs is the network being stretched, and no additional triggers can be exploited to enhance the mechanical strength.

Here, we describe a set of swelling-strengthening hydrogels (SSHs) achieved by virtue of a biological membrane barrier-inspired strategy (Fig. 1b, c). Liposomal membrane nanobarriers that are covalently embedded in a cross-linked network are used to regulate transmembrane transport. After swelling, the stretched network deforms the embedded membrane nanobarriers and subsequently initiates the transmembrane diffusion of the encapsulated molecules, which can trigger the formation of a new network from the preloaded precursor and form a double-network structure. Thanks to the tough nature of double-network hydrogels, SSHs demonstrate their swelling-strengthening behavior without the assistance of external stimuli and additional additives. Swelling-triggered self-strengthening enables the preparation of various dynamic materials, which may greatly expand their application in physiological environments.

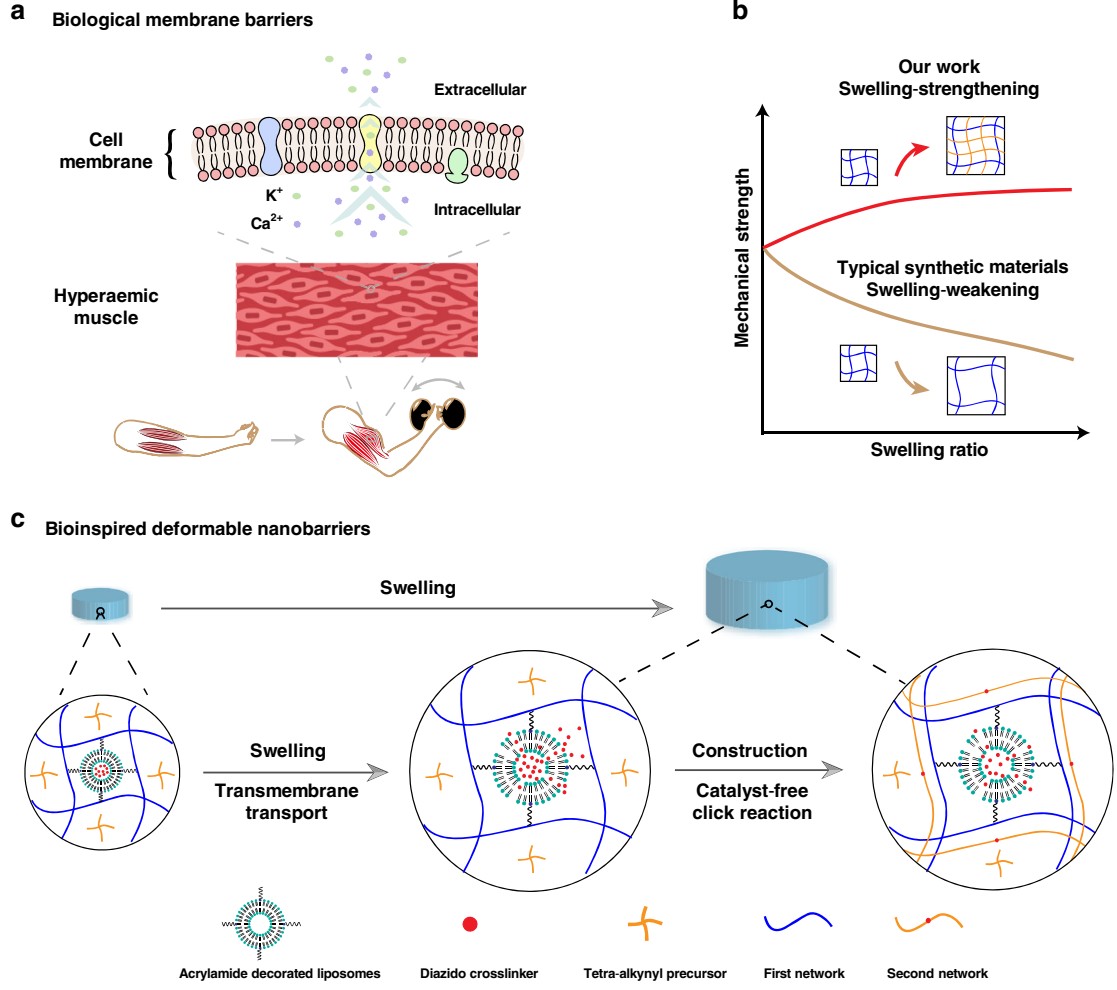

**Fig. 1 Schematic illustration of the swelling-strengthening behavior. a** Biological membrane barriers existed in living tissues, such as muscle. Transmembrane transport that is responsible for the hyperemia-mediated increasement of muscle stiffness depends on the ionic channel, transporters, diffusion, etc. **b** Variation of mechanical strength after swelling between typical synthetic materials and the swelling-strengthening hydrogels (SSHs) described in this work. **c** Development of SSHs via a biological membrane barrier-inspired strategy. Transport across the synthetic liposomal membranes mainly relies on diffusion.

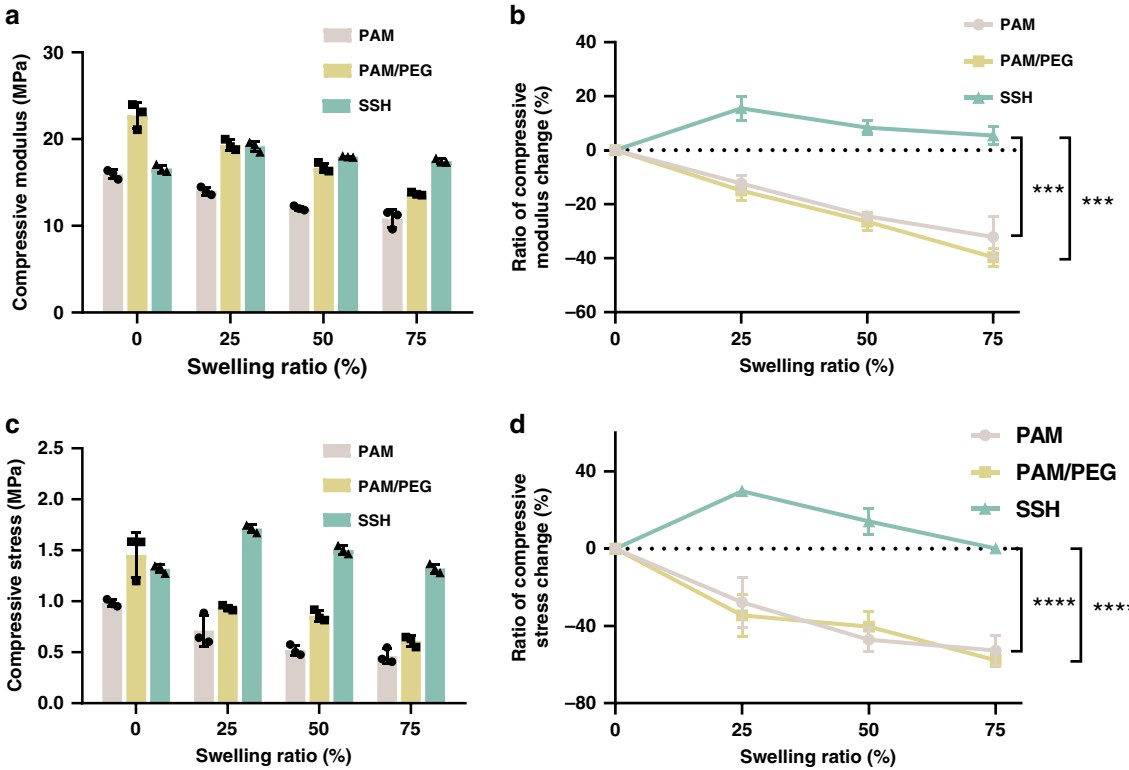

**Fig. 2 Swelling-strengthening behavior of swelling-strengthening hydrogels (SSHs) in vitro. a** Compressive modulus and **b** variation of compressive modulus versus swelling ratio. **c** Compressive stress and **d** variation of compressive stress at 90% strain versus swelling ratio. The strain rate was 5 mm/min, and the dimension of the cylindrical samples was 5 × 3 mm. All error bars represent the mean ± SD ($n = 3$ independent experiments). Significance was assessed using one-way ANOVA with Tukey's multiple comparisons test, giving $P$ values, ***$P < 0.001$, ****$P < 0.0001$.

## Results and discussion

**Design and preparation of SSHs**. To build an artificial yet robust system that possesses the features of natural membrane barriers, liposomes were embedded within hydrogel materials to simultaneously isolate trigger molecules and regulate their transmembrane transport[29–31]. Our strategy was shown in Fig. 1c. The liposomes decorated with double bonds were covalently incorporated into the first network by aqueous radical polymerization. The precursor of the second network was preloaded within the hydrogel, and the clickable crosslinker was prestored inside the liposomes, respectively. The swelling-triggered transmembrane transport of the crosslinker led to a catalyst-free yet highly efficient click reaction[32–34] between the preloaded precursor and the diffused crosslinker, forming the second polymer network. Consequently, the mechanical strength after swelling could exceed the initial SSHs when the increment of the strength resulted from the formation of the double-network structure transcended the strength loss caused by swelling[35–39]. As a proof-of-concept study, we chose water-soluble 1,11-diazido-3,6,9-trioxaundecane ($N_3$–$PEG_3$–$N_3$) and dibenzocyclooctyne end-capped 4-arm polyethylene glycol (Tetra-PEG-DBCO) as the clickable crosslinker and the precursor of the second network, respectively. Liposomes decorated with double bonds were prepared by co-assembly of hydrogenated soybean phospholipids with cholesterol and distearoyl phosphoethanolamine-$PEG_{2000}$-acrylamide, while Tetra-PEG-DBCO was synthesized via the amidation of DBCO acid with Tetra-PEG-amine (Supplementary Figs. 1 and 2). The decorated liposomes loaded with $N_3$–$PEG_3$–$N_3$ (Supplementary Fig. 3 and Supplementary Table 1) were covalently linked to the first network by aqueous radical copolymerization of acrylamide and $N,N'$-methylenebisacrylamide. SSHs were obtained by

preloading Tetra-PEG-DBCO inside the polyacrylamide (PAM) hydrogel (Supplementary Fig. 4).

**Swelling-strengthening behavior of SSHs**. The optimized SSHs showed excellent swelling-strengthening behavior. As shown in Fig. 2a, b and Supplementary Fig. 5, the compressive modulus of the SSH increased by 15.6% ± 4.5 (mean ± SD, three parallel samples) at a swelling ratio of 25%. Even with the swelling ratio increasing to 75%, the SSH could retain its initial mechanical strength. However, a typical swelling-weakening phenomenon was observed for the control hydrogels. The compressive moduli of both the corresponding PAM single-network and PAM/PEG double-network hydrogels decreased continuously during swelling, which declined by 32.1% ± 7.7 and 39.7% ± 3.4 at the swelling ratio of 75%, respectively. Further, the compressive stress of the SSH at 90% strain increased from 1.32 to 1.71 MPa as the swelling ratio increased by 25% (Fig. 2c, d). With the swelling ratio further increasing up to 75%, the compressive stress was able to maintain at the same level of the gel before swelling. As expected, the compressive stress of both control hydrogels at 90% strain reduced largely and showed a reduction of 52.8% ± 7.7 and 57.7% ± 3.2 at the swelling ratio of 75%, respectively. In addition, the compressive strength of the swelled PAM/PEG hydrogel was lower than that of SSH. This could be explained by the presence of a large number of crosslinked liposomes in SSH, which increased the mechanical strength of the gel.

SSHs were then subcutaneously implanted in the back of rats to investigate the swelling-strengthening behavior in a physiological environment (Fig. 3a, b and Supplementary Fig. 6). After implantation in the body, the SSHs were infiltrated by the body

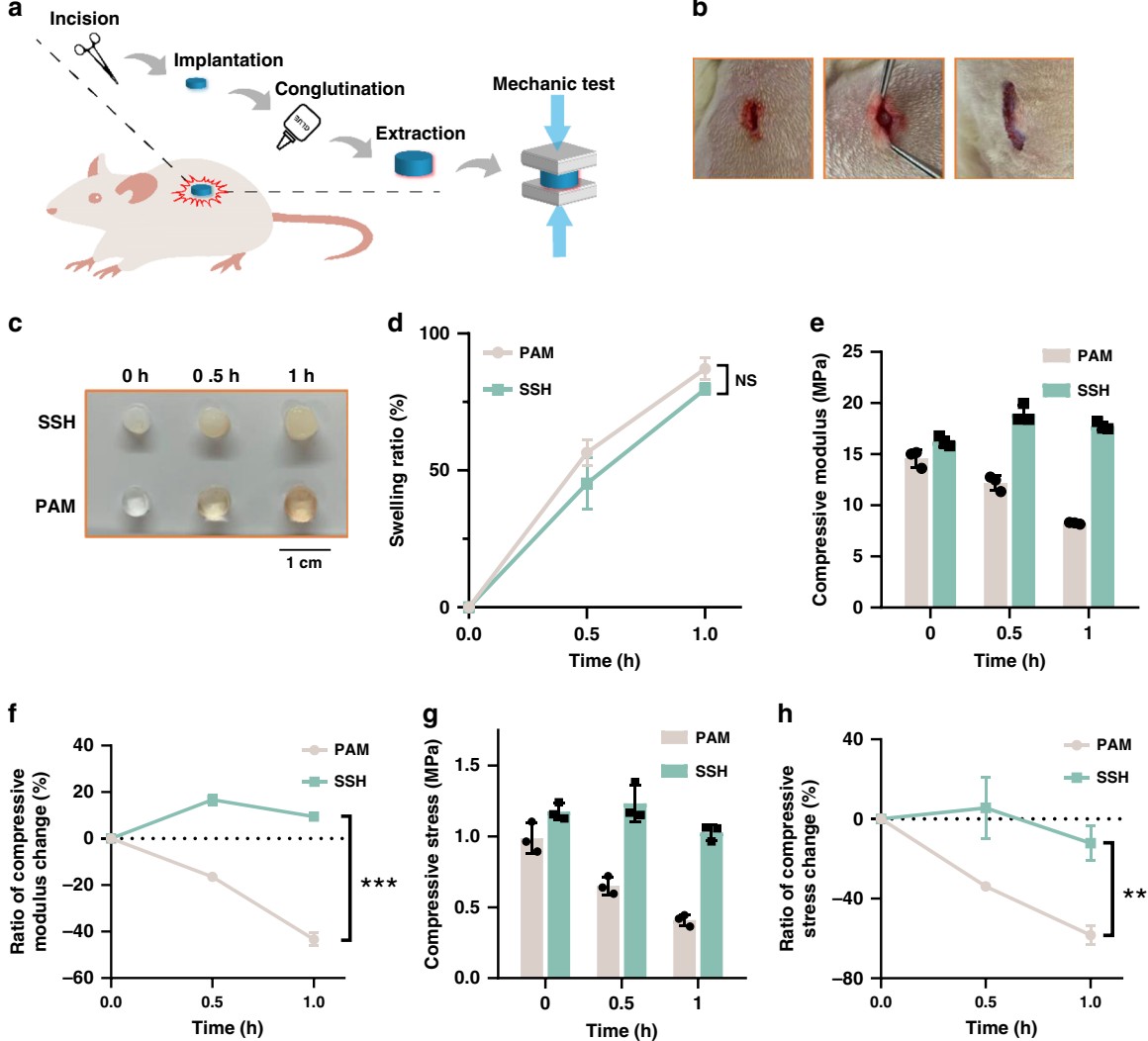

**Fig. 3 Swelling-strengthening behavior of swelling-strengthening hydrogels (SSHs) in vivo. a** Schematic diagram of the experimental procedure. **b** Photographs of the surgical implantation of SSHs in rats. Left: incision; middle: implantation; right: conglutination. A roughly 1-cm incision was made in the mediodorsal skin, and a lateral subcutaneous pocket was prepared. **c** Photographs and **d** swelling ratio of the hydrogel samples after implantation for the predetermined time points. **e** Compressive modulus at 90% strain versus swelling ratio. **f** Variation of compressive modulus versus swelling ratio. **g** Compressive stress at 90% strain versus swelling ratio. **h** Variation of compressive stress versus swelling ratio. The strain rate was 5 mm/min, and the dimension of the cylindrical samples was 5 × 3 mm. All error bars represent the mean ± SD ($n = 3$ independent experiments). Significance was assessed using unpaired two-tailed Student's $t$ test, giving $P$ values, **$P < 0.01$, ***$P < 0.001$. NS no significance.

fluids (Fig. 3c, d). Swelling ratios at the predetermined time points were calculated by measuring the weight change of the implanted sample. Similar to the PAM single-network hydrogel, the swelling ratio of the SSH increased by ~50% after implantation for 30 min, which further increased to ~85% with the time extending to 1 h. Remarkably, the compressive modulus of the SSH grew from $16.2 \pm 0.5$ to $18.9 \pm 0.9$ MPa ($P < 0.01$) after the swelling ratio approximated 50% (Fig. 3e, f). With the swelling degree further increasing to ~85%, the modulus could maintain at 1.09-fold higher than that of the gel before implantation. Oppositely, the compressive modulus of the control PAM gel lowered gradually after implantation, which dropped by $43.3\% \pm 2.8$ at the swelling degree of ~85%. Moreover, a similar tendency was observed for the corresponding compressive stress of SSHs at 90% strain (Fig. 3g, h), demonstrating the swelling-strengthening behavior under a complex in vivo condition. In addition, in vitro cell viability assay indicated negligible cytotoxicity of $N_3$–$PEG_3$–$N_3$ and Tetra-PEG-DBCO (Supplementary Fig. 7). The treated cells showed almost unaffected viability

even with the concentrations of $N_3$–$PEG_3$–$N_3$ and Tetra-PEG-DBCO increasing up to 1 and 60 mg/ml, respectively, which far exceeded their loading contents in SSH. The self-strengthening property of SSHs was further elucidated by mechanical deformation given that the stretched network could deform the embedded liposomal membrane nanobarriers. We speculated that the mechanical deformation of SSHs could similarly trigger the release of the loaded crosslinker and ultimately reinforce the strength. As displayed in Fig. 4a–d and Supplementary Figs. 8 and 9, both the tensile and compressive stress of the SSH climbed significantly after prestretching or precompressing, while the corresponding PAM single-network and PAM/PEG double-network hydrogels weakened apparently. The tensile stress of the SSH enhanced by $37.4\% \pm 16.5$ after prestretching to 100% its initial length for 5 min. Following prestretching to 200% its initial length, the tensile stress further increased by $51.7\% \pm 8.8$. In addition, the compressive stress of the SSH at 70% strain raised by $33.5\% \pm 11.6$ after compression to 70% its initial height for 5 min, which could enlarge by $47.9\% \pm 15.1$ with the loading time

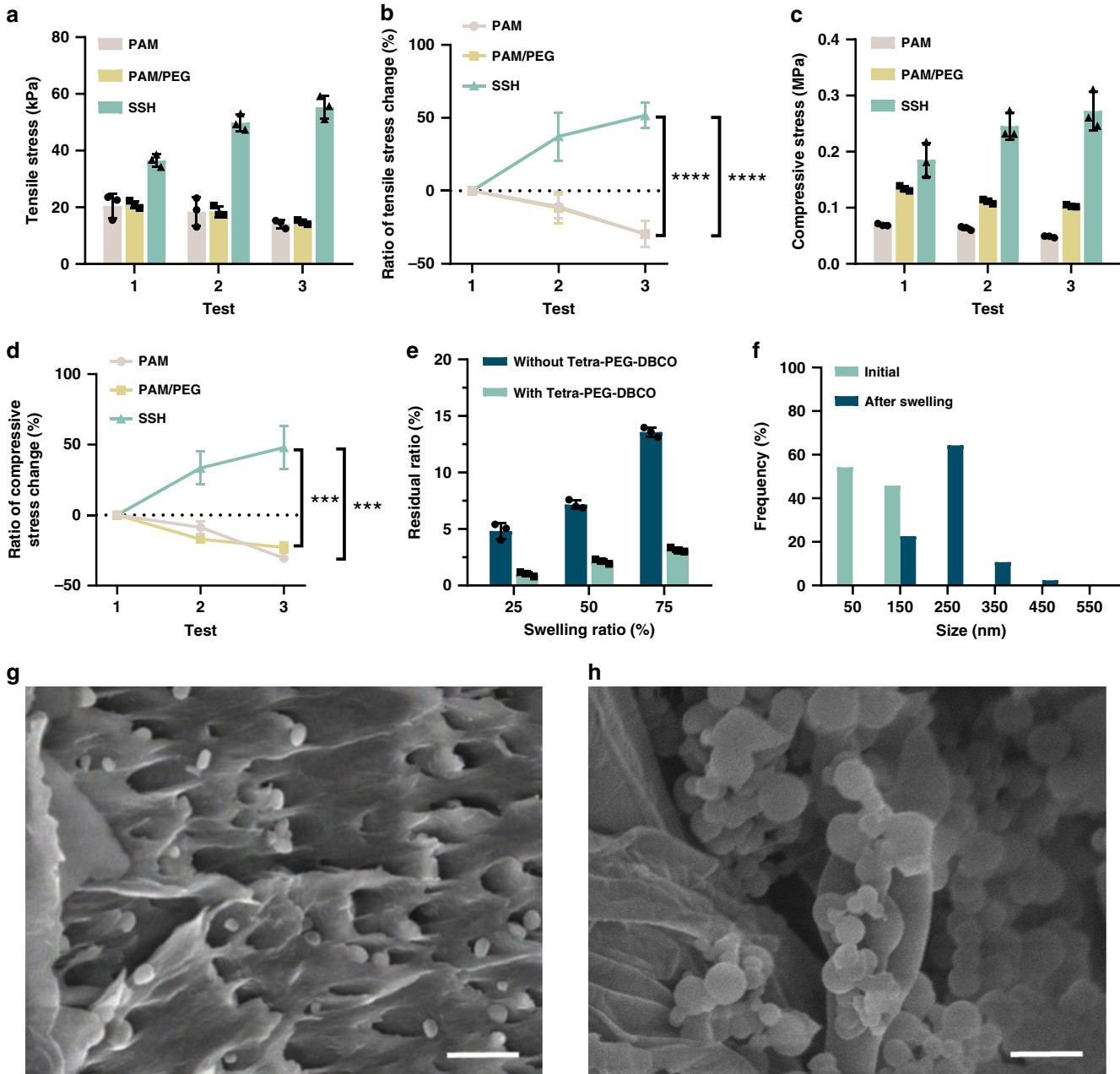

**Fig. 4 Swelling-triggered transmembrane transport and click reaction. a** Tensile stress and **b** variation of tensile stress under different tensile conditions. (1) direct test; (2) test after prestretching at 100% strain for 5 min; (3) test after prestretching at 200% strain for 5 min. **c** Compressive stress and **d** variation of compressive stress at 70% strain under different compression conditions. (1) direct test; (2) test after compression at 70% strain for 5 min; (3) test after compression at 70% strain for 10 min. **e** Relationship between released yet unreacted N3-PEG3-N3 and swelling ratio of the gels with or without Tetra-PEG-DBCO. **f** Average size of the liposomes embedded in the gels before and after swelling, respectively. Scanning electron microscopy (SEM) images of the lyophilized swelling-strengthening hydrogels (SSHs) (**g**) before and (**h**) after fully swelling. Scale bar, 500 nm. Each test was repeated three times independently with similar results. All error bars represent the mean ± SD ($n = 3$ independent experiments). Significance was assessed using one-way ANOVA with Tukey's multiple comparisons test, giving $P$ values, ***$P < 0.001$, ****$P < 0.0001$.

extending to 10 min. Similar strengthening was observed for both the tensile and compressive moduli of the SSH (Supplementary Fig. 10). In contrast, the weakening behavior of both control gels was disclosed under the same experimental condition. Representatively, the PAM gel appeared a maximal decrease of 29.8% ± 9.1 and 18.0% ± 12.9 in tensile stress and modulus as well as 30.5% ± 2.0 and 32.5% ± 0.9 in compressive stress and modulus, respectively. We further investigated the influence of the rate of deformation on the mechanical strength of SSH. Interestingly, the tensile stress of the gel increased with the decrease of the rate of

deformation, which could be attributed to the insufficient release of $N_3–PEG_3–N_3$ under a short period of time (Supplementary Fig. 11).

**Criteria required for swelling-strengthening behavior.** To realize swelling-strengthening behavior, the key challenge is to ensure that the mechanical strength of the eventually formed double-network structure is higher than that of the initial gel, which depends on the amount of the diffused crosslinker, the

content of the preloaded precursor, and the efficiency of the click reaction. Therefore, achieving swelling-strengthening requires three correlative effects. First, the crosslinker should be triggered to diffuse across the liposomal membrane nanobarriers efficiently by the stretched network after swelling. Second, the released crosslinker should react with the preloaded precursor via a highly efficient catalyst-free reaction. Lastly, the mechanical strength of the formed double-network structure should be optimized to exceed the initial single-network gel. The rational design of SSHs was on the basis of three criteria: (1) the liposomes must have high stability and limited leakage of the loaded crosslinker, while can be deformed to initiate rapid transmembrane transport; (2) the ratio of the diffused crosslinker to the preloaded precursor must be matched to form a crosslinking network by click reaction; (3) both the concentration and crosslinking density of the second network must be optimized to strengthen the ultimate gel.

We investigated the effect of liposomal composition on the stability and triggered transmembrane diffusion of the $N_3–PEG_3–N_3$ loaded liposomes. The liposomes optimized with a component of hydrogenated soybean phospholipids/cholesterol/distearoyl phosphoethanolamine-PEG$_{2000}$-acrylamide at a molar ratio of 50/45/5 exhibited extraordinary stability and had negligible leakage even with storage time increasing up to 30 days (Supplementary Fig. 12). The release of $N_3–PEG_3–N_3$ from the embedded liposomes was initiated by swelling and could increase with the swelling degree of the gel (Fig. 4e). The release ratio approached $4.8\% \pm 0.7$, $7.2\% \pm 0.4$, and $13.6\% \pm 0.4$ with the swelling degree increasing to 25%, 50%, and 75%, respectively. We speculated that the mechanoresponsive release of $N_3–PEG_3–N_3$ resulted from the stretched network, which enlarged the embedded liposomes and loosened the lipid bilayer arrangement after swelling. This hypothesis was verified by scanning electron microscopy (SEM) measurements. SEM images validated that the average size of the embedded liposomes increased from 97 to 230 nm after swelling (Fig. 4f–h). In contrast to dynamic light scattering (DLS) measurement, the decrease of a size determined by SEM could be simply attributed to the dehydration of the liposomes after lyophilization. The triggered release of $N_3–PEG_3–N_3$ was further confirmed by stretching, and the accumulated release could reach up to $30.9\% \pm 19.2$ after prestretching to 300% its initial length for 5 min (Supplementary Fig. 13). Similarly, the liposomes were deformed into a flat shape in the same direction that the gel was stretched (Supplementary Fig. 14). We then studied the click reaction by preloading with a given amount of Tetra-PEG-DBCO inside the gel. As illustrated in Fig. 4e, $N_3–PEG_3–N_3$ reacted with Tetra-PEG-DBCO upon release. The conversion of $N_3–PEG_3–N_3$ was determined by calculating the ratio of unreacted/released. Importantly, the efficiency of the click reaction remained consistent and could reach up to ~80% during swelling. In view of the high yield of the click reaction, the unattached polymer was statistically negligible as each polymer containing four DBCO reactive groups. A set of hydrogels with various concentrations of the second network were synthesized to further explore the effect of the double-network composition on their strength. All gels were transparent, and no phase separation was observed during the experiments (Supplementary Fig. 15e). This might be ascribed to the low polymer concentration of the gels, in which the total contents were <20 wt%. The compressive stress at 90% strain increased from 0.89 to 3.43 MPa, with the loading of the PEG network, increasing to 10 wt% (Supplementary Fig. 15). With the assistant of these correlative effects, the optimized SSH consisting of 4.5 wt % of Tetra-PEG-DBCO and a feed ratio of 22.4 μmol/ml of $N_3–PEG_3–N_3$ successfully achieved the swelling-strengthening behavior. Namely, the increment of the strength resulted from the formation of the double-network structure exceeded the loss of

strength at a swelling degree of 75%. Taken together, on the basis of the kinetics of both the transmembrane diffusion and the click reaction at a given swelling degree, the mechanical strength of SSHs after swelling could be regulated to accomplish the swelling-strengthening phenomenon by adjusting the loading of the second network.

**Verifying the formation of the second network**. To further confirm the structure of the second network responsible for the mechanical behavior of SSHs, we designed and synthesized a triggerable PAM network that could enable a comprehensive characterization of the newly formed network (Fig. 5a). Namely, the first network was replaced by a disulfide-crosslinked PAM network, which was reduction-responsive. Therefore, the initial network could be dissolved by glutathione (GSH), a reducing agent, after the formation of the double-network structure. Once the first network was removed, the newly formed network could be obtained. As shown in Fig. 5b, the disulfide-crosslinked PAM network was triggered to dissolution by GSH. As expected, the swelled hydrogel could retain its initial solid shape after treating with GSH (Fig. 5b–e), verifying the formation of the second network. The purified second network was obtained by complete removal of the de-crosslinked PAM via diffusion. Mechanical testing showed that the formed second network had a tensile stress of 28.4 kPa at 600% strain and compressive stress of 915 kPa at 90% strain, respectively (Fig. 5f, g). SEM images evidenced the morphologic difference between the double-network and the second network hydrogels (Fig. 6a, b). Peaks at 61.4 ppm (assigned to $–NCH_2CH_2O–$) and 57.7 ppm (assigned to $–NCH_2CH_2O–$) in the solid-state $^{13}C$ nuclear magnetic resonance ($^{13}C$ NMR) spectrum reflected that the second network was crosslinked by $N_3–PEG_3–N_3$ via click reaction (Fig. 6c). Meanwhile, in comparison with the solid-state $^{13}C$ NMR spectrum of the PAM network[40], the disappearance of peaks around 183.1 ppm (assigned to $C = O$) and 42.3 ppm (assigned to $–CH–CH_2–$) suggested the removal of the primary network. Absorbances of the ring stretching and plane bending of 1,2,3-triazole unit at 1450, 1250, 1050, and 750 cm$^{-1}$ in the Fourier-transform infrared (FTIR) spectrum further demonstrated the crosslinking of the precursor by click reaction (Fig. 6d). The absence of a signal at 3340 and 3168 cm$^{-1}$ that was assigned to the N–H stretching of the amide group in the PAM network further demonstrated the disappearance of the primary network (Supplementary Fig. 16)[41]. The absorbance peak at 3400 cm$^{-1}$ revealed that the two networks were intertwined by hydrogen bonds. These results well confirmed the mechanical behavior, microscopic morphology, and molecular structure of SSHs.

In summary, we report the swelling-strengthening behavior of synthetic hydrogel materials achieved by a biological membrane barrier-inspired strategy. The designed SSHs automatically switch from a single-network to a double-network structure via a catalyst-free click reaction without the help of external triggers and present increased mechanical strength after swelling. Central to the swelling-strengthening nature of SSHs is the existence of many artificial biological membrane nanobarriers that enable a swelling-triggered transmembrane transport. In addition to this swelling-strengthening property, other spontaneous and dynamic behaviors such as swelling-induced destroying, color-switching, or biological reaction could be incorporated into synthetic hydrogels by the isolation of the corresponding trigger molecules, studies that are currently ongoing.

## Methods
**Materials**. Hydrogenated soybean phospholipids (HSPC), cholesterol (Chol), and distearoyl phosphoethanolamine-PEG$_{2000}$-acrylamide (DSPE-PEG$_{2000}$-ACA) were purchased from Shanghai Ponsure Biotech, Inc. Four-arm poly(ethylene glycol)

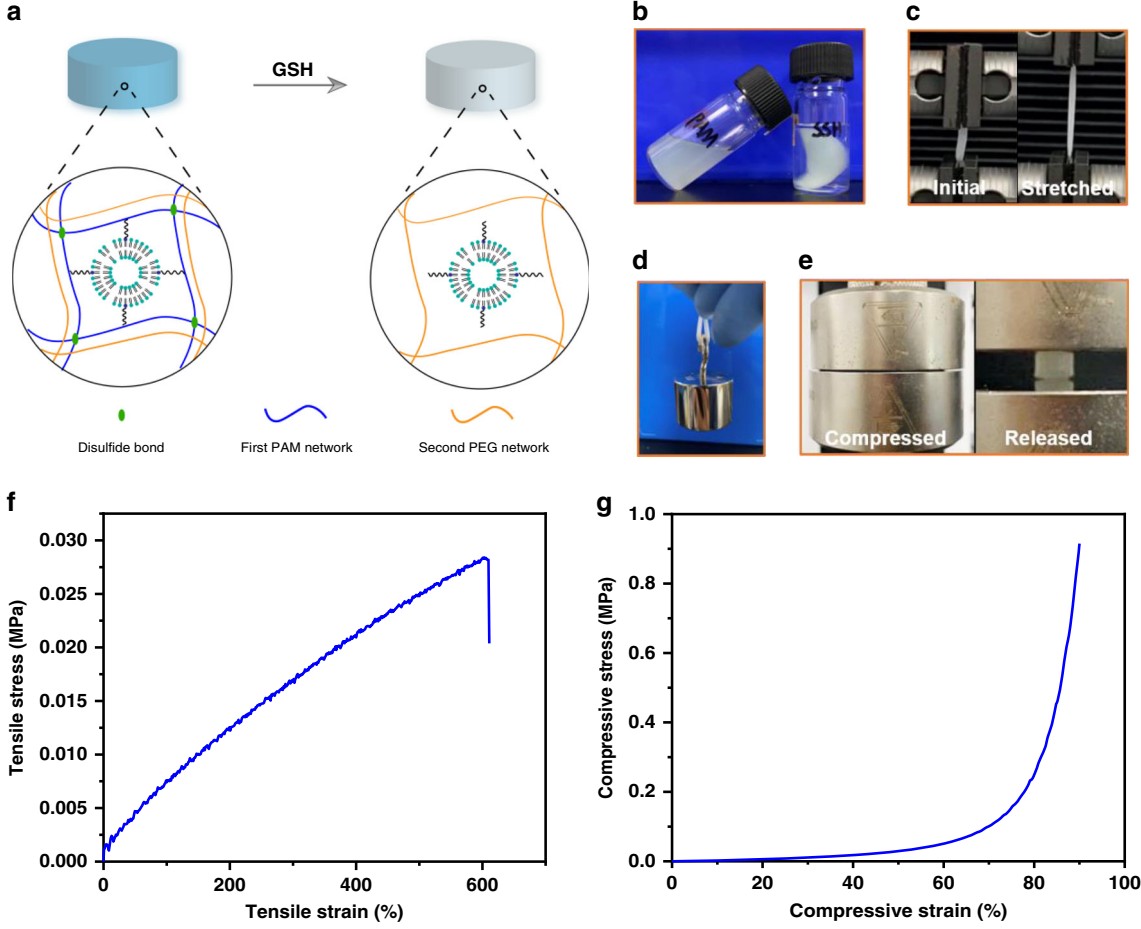

**Fig. 5 Design, preparation, and mechanical testing of the second network. a** Schematic illustration for the preparation of the second network via GSH-triggered dissolution of the disulfide-crosslinked polyacrylamide (PAM) network. The reduction-responsive PAM network was synthesized by using $N,N'$-bis(acryloyl)cystamine as a crosslinker. **b**–**e** Photographs of different samples. **b** PAM single-network hydrogel (left) and triggerable swelling-strengthening hydrogels (SSH) (right) samples after soaking in 250 mM GSH for 1 day. **c** A triggerable SSH strip stretched to its maximal tensile strain. **d** A triggerable SSH strip ($25 \times 4 \times 1.5$ mm) bearing a load of 100 g. **e** A triggerable SSH cylinder compressed to 90% strain and then released. Hydrogel samples were fully hydrated throughout the experiment. Curves of (**f**) tensile and (**g**) compressive stress versus strain of the purified second network hydrogels.

amine (Tetra-PEG-amine) was provided by Xiamen Sinopeg company. Dibenzocyclooctyne-acid (DBCO acid, ≥95%) was supplied by Xi'an ruixi Biological Technology Co., Ltd. O-(7-azabenzotriazol-1-yl)-N,N,N',N'-tetra-methyluronium hexafluorophosphate (HATU, 99%) and glutathione (GSH, 99%) were purchased from J&K Scientific Ltd. N,N-diisopropylethylamine (≥99%, DIEA), N,N'-methylenebisacrylamide (99%, MBAA), N,N,N',N'-tetra-methylethylenediamine (99%, TEMED), and anhydrous N,N-dimethylformamide (99.8%, DMF) were provided by Sigma. Ammonium persulfate (99.99%, APS) and 1,11-diazido-3,6,9-trioxaundecane (≥95%, $N_3$-$PEG_3$-$N_3$) were supplied by Aladdin. Acrylamide (99%, AM) was purchased from Adamas. Ethyl ether (≥99.7%, $Et_2O$) and chloroform (≥99%, $CHCl_3$) were provided by Sinopharm chemical reagent Co., Ltd. Ammonium sulfate (≥99%, $(NH_4)_2SO_4$) and PBS buffer (10×, pH 7.4) were supplied by Sangon Biotech company. The PBS buffer used in all experiments was diluted ten times. Deionized water was used to prepare all aqueous solutions. Other reagents and solvents were used as received.

**Synthesis and characterization of Tetra-PEG-DBCO**. The preloaded precursor Tetra-PEG-DBCO was synthesized by a one-step amidation reaction between DBCO acid and Tetra-PEG-amine[42] (Supplementary Fig. 1). DBCO acid (171 mg, 0.56 mmol), HATU (213 mg, 0.56 mmol), and 3 ml anhydrous DMF were added into a dry flask to give a light-yellow solution under stirring. Tetra-PEG-amine ($M_n$ = 40 kDa, 2.8 g, 0.07 mmol) in 12 ml of anhydrous DMF was then added at 0 °C. After the addition of 200 μL of anhydrous DIEA, the reaction mixture was stirred at 25 °C for 36 h. Afterward, the mixture was dropwise added into cold $Et_2O$ with stirring to afford a white solid powder. The product was further washed by $Et_2O$ for three times and dried under vacuum to give a light-yellow powder (10.5 g, yield 91.3%).

The structure of the obtained compound was confirmed by proton nuclear magnetic resonance ($^1$H NMR) spectrum (Bruker Avance 500 spectrometer). Chemical shifts (δ) were expressed in ppm. According to 1H NMR (Supplementary

Fig. 2), the end-functionalization was 91.2%. $^1$H NMR (400 MHz, CDCl₃): 7.67–7.32 (m, 32H), 6.19 (br, 4H), 5.17 (d, J = 13.6 Hz, 4H), 3.86–3.31 (m, 3678H), 2.80–1.93 (m, 16H) ppm.

**Preparation of NALip**. The $N_3$–$PEG_3$–$N_3$-loaded acrylamide decorated liposomes (NALip) were prepared via the conventional thin-film hydration method[43]. Briefly, HSPC/Chol/DSPE-$PEG_{2000}$-ACA (molar ratio: 50/45/5) was added in a round-bottom flask and dissolved by $CHCl_3$. Lipid films were acquired after removing $CHCl_3$ by rotary evaporation. Then the lipids were hydrated by $(NH_4)_2SO_4$ aqueous solution (155 mM) at 50 °C under ultrasonic vibration for 20 min. The generated white solution was extruded through polycarbonate membranes (pore size: 200 nm) with the help of a mini extruder (Avavti Polar Lipids, USA) to homogenize the size of liposomes[44]. Acrylamide decorated liposomes (ALip) were obtained after removing $(NH_4)_2SO_4$ solution through ultrafiltration and further dispersed in 1× PBS solution.

$N_3$–$PEG_3$–$N_3$ was incubated together with the obtained ALip in an $N_3$–$PEG_3$–$N_3$/HSPC molar ratio of 3/1 at 56 °C for 20 min. Finally, the resultant NALip was purified by repeated ultrafiltration to remove the unencapsulated $N_3$–$PEG_3$–$N_3$, and further decentralized into 1× PBS solution.

**Morphology and size measurements of ALip**. The morphology of ALip was observed by transmission electron microscopy (TEM, HT7700 Exalens, Hitachi, Japan) at an accelerating voltage of 10 kV. In total, 10 μL of ALip dilute solution was dripped on the surface of a carbon-coated copper grid (400 mesh). After deposition for 5 min, the excess solution was sucked away by filter paper. Then ultrapure water was repeatedly added to the surface of the grid and absorbed by filter paper to remove the residual salts. The obtained grid was negatively stained by uranyl acetate solution (1 wt%, 5 μL) and dried completely prior to the test. The size and poly-dispersity index (PDI) of ALip were determined by DLS (Malvern Zetasizer Nano

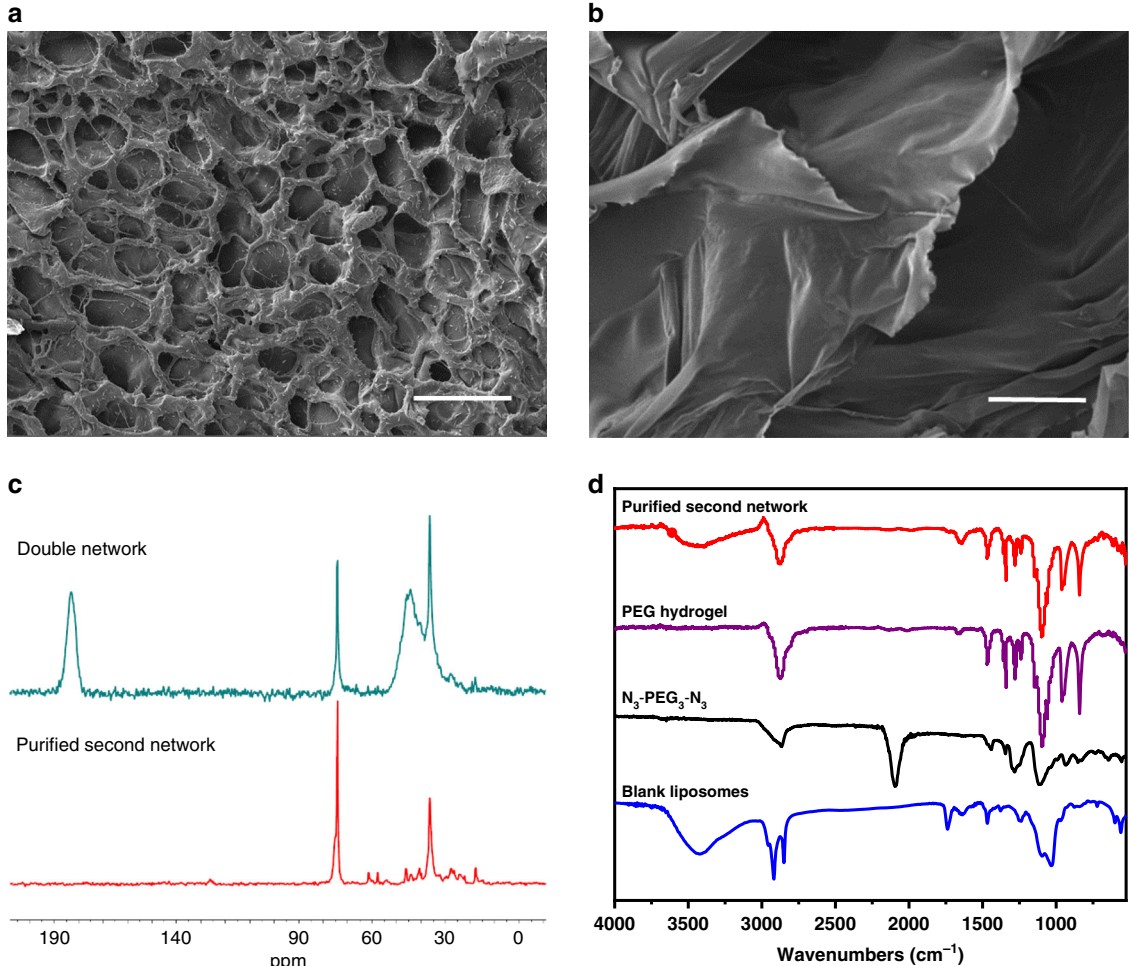

**Fig. 6 Physical characterization of the second network.** Scanning electron microscopy (SEM) images of the triggerable swelling-strengthening hydrogels (SSHs) (**a**) before and (**b**) after treating with GSH. Samples were lyophilized before SEM measurement. Scale bar, 50 μm. Each test was repeated three times independently with similar results. **c** Solid-state 13C NMR spectra of the triggerable SSHs and the purified second network. The samples were lyophilized before analysis. **d** Fourier-transform infrared (FTIR) spectra of N3-PEG3-N3, blank liposomes, PEG hydrogel, and the purified second network.

ZSP). In all, 2 ml of ALip dilute solution was added into a quartz cuvette for the DLS measurements. Samples were tested in parallel for three times. TEM (Supplementary Fig. 3a) and DLS (Supplementary Fig. 3b) results show that the prepared ALip presented a uniform size with an average diameter of 173.9 ± 2.01 nm.

**Quantification of entrapment efficiency of $N_3$–$PEG_3$–$N_3$ in NALip.** The entrapment efficiency of $N_3$–$PEG_3$–$N_3$ was evaluated in the virtue of ultraviolet (UV) spectrophotometer. The standard curve of absorbance versus concentration of $N_3$–$PEG_3$–$N_3$ was obtained at an absorption wavelength of 278 nm (Supplementary Fig. 3c). After incubation with ALip, a portion of $N_3$–$PEG_3$–$N_3$ was encapsulated in Alip to acquire NALip, unencapsulated $N_3$–$PEG_3$–$N_3$ was completely transferred to the subnatant by ultrafiltration. The amount of unencapsulated $N_3$–$PEG_3$–$N_3$ in the subnatant can be calculated according to the standard curve. Finally, the encapsulated ratio and loading amount of $N_3$–$PEG_3$–$N_3$ in NALip can be obtained. The characteristic peaks of $N_3$–$PEG_3$–$N_3$, lyophilized NALip were recorded via FTIR spectroscopy to verify the successful encapsulation of $N_3$–$PEG_3$–$N_3$ in ALip (Supplementary Fig. 3d). The molar feed ratio of $N_3$–$PEG_3$–$N_3$ to HSPC showed a significant impact on the entrapment efficiency of $N_3$–$PEG_3$–$N_3$ in the process of incubation. The loading amount and encapsulation efficiency of $N_3$–$PEG_3$–$N_3$ in NALip under different feed ratios are listed in Supplementary Table S1. The optimized ratio of 3:1 was employed in the preparation of NALip for the following experiments.

**Preparation of SSHs, PAM, and PAM/PEG hydrogels.** Using concentrated NALip solution as a solvent, AM (284 mg, 4 mmol), crosslinker MBAA (40 μL, 0.1 M, 4 μmol), initiator APS (5‰ w/w), catalyst TEMED (5‰ w/w), and Tetra-PEG-DBCO were dissolved to give a pre-gel solution at room temperature in a nitrogen glove box. After 30 min polymerization in PTFE molds, SSHs were obtained and ready for further measurements. Noted that the amount of Tetra-PEG-DBCO depended on the encapsulation efficiency and released ratio of

$N_3$–$PEG_3$–$N_3$ in NALip. Tetra-PEG-DBCO was added in the stoichiometric proportion (molar ratio, 1:1) according to the release ratio of $N_3$–$PEG_3$–$N_3$. Both PAM and PAM/PEG hydrogels were similarly prepared as controls. Briefly, PAM was prepared by mixing AM, MBAA, APS, and TEMED in aqueous solution, and PAM/PEG was synthesized via blending AM, MBAA, APS, TEMED, Tetra-PEG-DBCO, and free $N_3$–$PEG_3$–$N_3$ in ultrapure water. The photographs of the above hydrogels are shown in Supplementary Fig. 4.

**Swelling ratio (SR) of the hydrogels.** Hydrogels were weighed immediately after polymerization to determine the weight ($W_D$), and then incubated in a pre-determined amount of ultrapure water ($W_H$) at 37 °C for absorption. A period of 8 h was taken to reach a complete swelling ratio of 25, 50, and 75%. SR of the hydrogels was calculated according to the following Eq (1)[45].

$$SR = W_H / W_D. \qquad (1)$$

**In vivo swelling experiments.** All the animal procedures complied with the guidelines of the Shanghai Medical Experimental Animal Care. Animal protocols were approved by the Institutional Animal Care and Use Committee of Shanghai Jiao Tong University School of Medicine. Male rats (250–300 g) were used for in vivo swelling studies. A roughly 1-cm incision was made in the mediodorsal skin, and a lateral subcutaneous pocket was prepared. The SSH and PAM samples ($n = 3$; 5 × 3-mm cylinders) were implanted inside, and the incision was bonded with the help of tissue glue under sterile conditions. At designated time intervals (0.5 h and 1 h), the rats were sacrificed, and then the implanted samples were retrieved for the subsequent compression tests.

**In vitro cell viability assay.** A cell counting kit-8 (CCK-8, Beyotime Institute of Biotechnology, China) was used to investigate the cytotoxicity of $N_3$–$PEG_3$–$N_3$ and

Tetra-PEG-DBCO against 4T1 cells. Briefly, cells were seeded at a density of $1 \times 10^4$ cells per well in a 96-well plate, and cultured overnight in RMPI 1640 (Hyclone, Thermo Scientific) supplemented with 10% fetal bovine serum (Hyclone, Thermo Scientific) and 1% penicillin & streptomycin (GIBICO, Invitrogen) at 37 °C in a humidified atmosphere containing 5% $CO_2$. Subsequently, 100 μl of fresh medium containing $N_3$–$PEG_3$–$N_3$ at different concentrations (100, 500, or 1000 μg/ml) or Tetra-PEG-DBCO at different concentrations (6000, 10000, or 60000 μg/ml) was added to replace the culture medium for 24 and 48 h, respectively. After incubation, the culture medium containing $N_3$–$PEG_3$–$N_3$ or Tetra-PEG-DBCO was removed, 100 μl of fresh medium and 10 μl of CCK-8 solution were added to each well. After incubation for 3 h, the OD value of cultures was recorded at 450 nm with the help of a Multi-Detection Microplate reader (BioTek, USA). The well with medium and CCK-8 solution but without cells, $N_3$–$PEG_3$–$N_3$, and Tetra-PEG-DBCO was used as a blank group. The well with medium, cells, and CCK-8 solution but without $N_3$–$PEG_3$–$N_3$ and Tetra-PEG-DBCO was employed as a control group. The experiment was replicated for four parallel samples.

**Mechanical characterization**. All the mechanical measurements were conducted on the universal material testing machine (Instron-3342, 50 N sensor) and repeated three times. Compression tests after swelling: The compression tests were carried out for samples with different swelling ratios. The strain rate was set at 5 mm/min, and the dimension of the cylindrical samples was $5 \times 3$ mm. The compression strain was set at 90%[46]. Tensile tests: The tensile properties were evaluated under three different tensile conditions. Test 1 was conducted directly. Test 2 was carried out after prestretching the sample at 100% strain for 5 min. Test 3 was initiated after prestretching the hydrogel at a strain of 200% for 5 min. The strain rate was set at 20 mm/min, and the dimension of the rectangle samples was $25 \times 4 \times 1.5$ mm. Repeated compression tests: Three compression tests were carried out on an identical sample under different compression conditions. Test 1 was conducted directly. Test 2 was carried out after compression at 70% strain for 5 min. Test 3 was initiated after compression at 70% strain for 10 min. Cyclic tensile tests: Cyclic tensile tests were conducted to 250% strain at different tensile rates (5, 50, and 200 mm/min), and the loading and unloading processes performed at the same rate. The tests were performed immediately following the initial loading for three times.

**SEM analysis**. The gross morphology of lyophilized SSH was observed via the Raman imaging combined with emission scanning electron microscopy (RI-SEM, TESCAN-MAIA3) under an accelerating voltage of 5 kV. The samples were completely swelled or stretched to 3 times their initial length before lyophilization. All the samples were gold-sputtered prior to the tests.

**Stability of NALip**. The stability of NALip was evaluated by detecting the leakage of $N_3$–$PEG_3$–$N_3$ encapsulated in NALip. The concentrated NALip solution (0.0224 μM) was stored in the dark at 4 °C for 30 days and then diluted by $1 \times$ PBS for ultrafiltration. The amount of $N_3$–$PEG_3$–$N_3$ in the subnatant was calculated by a UV spectrophotometer. There was no precipitation at the bottom of the vessel on the 30th day, and no $N_3$–$PEG_3$–$N_3$ was detected in the subnatant.

**Quantification of the transmembrane transport and click reaction**. Swelling-triggered transmembrane transport: The sample without Tetra-PEG-DBCO was employed to determine the transmembrane transport of the encapsulated $N_3$–$PEG_3$–$N_3$. The hydrogels were put in a given mass of ultrapure water at 37 °C to reach different swelling ratios. Then they were immersed in 1 ml of ultrapure water for 10 min. The concentration of $N_3$–$PEG_3$–$N_3$ in the incubation solution was determined with the help of the UV standard curve. Swelling-triggered click reaction: The sample with Tetra-PEG-DBCO was utilized to demonstrate that catalyst-free click reaction occurred between the released $N_3$–$PEG_3$–$N_3$ and the preloaded Tetra-PEG-DBCO. The experimental procedure was similar to the above Swelling-triggered transmembrane transport test. Tensile-triggered transmembrane transport: The sample without Tetra-PEG-DBCO was used to determine the transmembrane transport of $N_3$–$PEG_3$–$N_3$ under different stretching strain and time. The samples were immersed in 1 ml of water for 10 min after stretching, then the concentration of the released $N_3$–$PEG_3$–$N_3$ was measured by three parallel tests.

**Preparation and characterization of triggerable SSHs**. A disulfide crosslinker, $N,N'$-bis(acryloyl)cystamine, was synthesized via nucleophilic substitution reaction between cystamine dihydrochloride and acryloyl chloride according to the reported method[47]. Triggerable SSHs were synthesized by using $N,N'$-bis(acryloyl)cystamine as a crosslinker. The resulted SSHs were first incubated in a predetermined amount of ultrapure water at 37 °C until the samples were swelled completely, followed by soaking in 250 mM GSH solution for ~3 days. After the dissolution of the reduction-responsive PAM network, the newly formed PEG network was purified by diffusion. The obtained samples were tailored to strips ($20 \times 6 \times 2.5$ mm) and cylinders ($7 \times 4$ mm) for mechanical testing. Lyophilized hydrogel samples were used for SEM, FTIR, and solid-state $^{13}C$ NMR (Bruker AVANCE NEO, 600 MHz WB) measurements.

**Statistical analysis**. All error bars represent the mean ± s.d. A significant difference was assessed by Student's $t$ test between two groups or one-way analysis of variance with Tukey's multiple comparisons test among three groups, giving $P$ values, $*P < 0.05$, $**P < 0.01$, $***P < 0.001$ and $****P < 0.0001$. NS no significance.

**Reporting summary**. Further information on research design is available in the Nature Research Reporting Summary linked to this article.

## Data availability
All data presented in the paper are available from the authors upon reasonable request.

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

## Acknowledgements

This work was financially supported by the National Natural Science Foundation of China (21875135), the Recruitment Program of Global Youth Experts of China (D1410022), the Shanghai Municipal Education Commission-Gaofeng Clinical Medicine Grant Support (20181704, 20191820), and the Innovative research team of high-level local universities in Shanghai (SSMU-ZLCX20180701).

## Author contributions

J.L. and Y.P. conceived and designed the experiments. F.W. performed all experiments. All authors analyzed and discussed the data. F.W. and J.L. wrote the paper.

## Competing interests

The authors declare no competing interests.
