## [Peer Review File · Nature Communications]

Reviewers' Comments:

Reviewer #1:

Remarks to the Author:

The research article by Wu et al. entitled, "Swelling-strengthening hydrogels by embedding with deformable nanobarrriers" submitted to Nature Communications was originally submitted to Nature Nanotechnology and its revised version was submitted to Nature Communications. Despite the lengthy comments by the authors, the revision seems rather superficial and incomplete. The manuscript still needs a major revision for publication in this leading journal.

1. English. Even though the authors claim they have taken care of this issue, there are too many grammar errors throughout the manuscript. It is very surprising that the authors could not clean up grammar/stylistic errors in the resubmitted version.
2. Regarding the references on biological glues and artificial tissues, the authors added one recent review paper, which does not really talk about the dynamic nature of the hydrogel. The rest of references is irrelevant to biological glues. If the authors cannot find references on this topic, maybe they should rethink the application of this hydrogel and add related references.
3. Statistical tests are not rigorous. The authors explained that they used the t-test because they were comparing just two groups among the three. However, this is exactly the reason that various statistical correction methods were developed. The authors are still comparing three groups, although the comparison involved a common reference group. There are separate tests for such cases. And whenever a quantitative comparison is made, statistical significance should be provided. For example, when the authors stated, "the compressive modulus remained 1.05-fold higher than that of the initial gel." What was the statistical significance? It is misleading when 1.05 fold increase actually means there was no change.
4. The authors revised the sentences about "stress relaxation". However, the explanation they added is insufficient. They basically changed the term from "stress relaxation" to "viscoelasticity". What is the basic physics involved in this phenomenon? The authors must demonstrate that they themselves understand the physics of it instead of just adding related references.
5. The SEM image of "stretched" liposomes is much less convincing than before. What is shown in Figure S13 seems like some wrinkles of polymers or air pockets, not the liposomes. The authors also claimed that the size of liposomes increased after stretching. How is it possible? Shouldn't the volume of the liposomes remain the same and only the morphologies change? And how many SEM images were used for the size distribution in Figure S13? It looks like it was obtained based on one SEM image which is not convincing. And it appears that the large-sized liposomes the authors counted are just larger wrinkles.
6. When the authors showed a picture of a 100g weight hanging on the hydrogel, does it match their tensile strength measurement? What was the dimension of the hydrogel that was used for the experiment? Was the hydrogel still fully hydrated?

Reviewer #2:

Remarks to the Author:

Article by Wu et al. reported that mechanical strength of hydrogels can be enhanced at higher swelling ratio because reactive precursors in vesicle can be released by swelling-induced stretching and form new polymer network. It is interesting to incorporate vesicles containing clickable monomers into hydrogel network for enhancing their mechanical property when it was stretched upon swelling without further treatment.

However, as authors mentioned already in the main text, Gong et al. (ref. 27) reported swelling- or stretching-induced double network formation inside hydrogel, in which swelling-strengthen phenomena can be even further continued upon external stimuli just like muscle. By contrast, the

hydrogels in this report was strengthened upon swelling once but not any more than once.

Therefore, I believe that authors need to justify why this single-use swelling-strengthening hydrogels should be developed or find proper application. Swelling-strengthening phenomenon itself would not be justification for publication in this journal because there are many examples of stronger hydrogels in swollen state including *Polymers* 2018, 10, 1025 (doi:10.3390/polym10091025).

But, again, this work may be well received in specialized journal such as biomaterials or Advanced healthcare materials. It would be interesting to encapsulate tetrafunctional monomers in separate vesicle.

Reviewer #4:

Remarks to the Author:

The authors have adequately addressed the concerns raised by Reviewer 3 in the first review. The method presented is interesting and solves the issue that hydrogels weaken with swelling. There are applications for this work in biological applications, which are supported by the biocompatibility results presented in the article.

I will reiterate that the clarity of the manuscript should be improved. For example:

- In the first sentence, "being" should be "are".
- Line 32, "hyperaemia" should be defined or omitted. For example, the sentence could be written: "For example, muscles become much stronger after swelling with blood (6,7)."
- It is not clear how hyperaemia swelling leads to increased stiffness. Instead, the mechanism of hyperaemia swelling itself is emphasized (e.g., Lines 32-37). This is akin to describing the swelling process of synthetic gels without mentioning that the swelling process weakens the gel by diluting the network. Why does muscle tissue not swell-weaken like synthetic hydrogels? The polymer network (e.g., collagen fibers) similarly dilutes upon swelling. Understanding this mechanism is important to the method presented in this paper.
- Line 36, "increasement" should be "increase".
- Line 54, "initiate" should be "initiates".
- Line 152, "highly efficient via a catalyst-free reaction" could instead be "via a highly efficient catalyst-free reaction".
- Line 204, "deigned" should be "designed".

Reviewer #1

The research article by Wu et al. entitled, "Swelling-strengthening hydrogels by embedding with deformable nanobarrier" submitted to Nature Communications was originally submitted to Nature Nanotechnology and its revised version was submitted to Nature Communications. Despite the lengthy comments by the authors, the revision seems rather superficial and incomplete. The manuscript still needs a major revision for publication in this leading journal.

Response: We thank the reviewer very much for taking his/her time to review our manuscript and help us to improve the quality of the work. The manuscript has been majorly revised according to his/her suggestions and requests.

1. English. Even though the authors claim they have taken care of this issue, there are too many grammar errors throughout the manuscript. It is very surprising that the authors could not clean up grammar/stylistic errors in the resubmitted version.

Response: According to the reviewer's request, the English writing has been polished further by proof-reading. We have tried our best to correct the grammar/stylistic errors in the revised manuscript.

2. Regarding the references on biological glues and artificial tissues, the authors added one recent review paper, which does not really talk about the dynamic nature of the hydrogel. The rest of references is irrelevant to biological glues. If the authors cannot find references on this topic, maybe they should rethink the application of this hydrogel and add related references.

Response: We thank the reviewer for highlighting this point. More relevant literatures (Ref. 28-33) that have demonstrated the need of dynamic nature of hydrogels for the applications in biological glues and artificial tissues have been cited in the revised manuscript.

3. Statistical tests are not rigorous. The authors explained that they used the t-test because they were comparing just two groups among the three. However, this is exactly the reason that various statistical correction methods were developed. The authors are still comparing three groups, although the comparison involved a common reference group. There are separate tests for such cases. And whenever a quantitative comparison is made, statistical significance should be provided. For example, when the authors stated, "the compressive modulus remained 1.05-fold higher than that of the initial gel." What was the statistical significance? It is misleading when 1.05 fold increase actually means there was no change.

Response: We agree with the reviewer that various statistical methods have been developed for different situations. The statistical analysis was reperformed in the revised version. Significant difference was assessed by Student's t-test between two groups or one-way analysis of variance with Tukey's multiple comparisons test among three groups (Figure 2, 3 and 4). The statement "the compressive modulus remained 1.05-fold higher than that of the initial gel" was meant to emphasize that even with the swelling ratio increased up to 75%, the

SSHs could remain its initial mechanical strength, while the control hydrogels exhibited a typical swelling-weakening phenomenon. Accordingly, the statement was modified to “Even with the swelling ratio increased up to 75%, the SSH could remain its initial mechanical strength” in the revised manuscript.

4. The authors revised the sentences about “stress relaxation”. However, the explanation they added is insufficient. They basically changed the term from “stress relaxation” to “viscoelasticity”. What is the basic physics involved in this phenomenon? The authors must demonstrate that they themselves understand the physics of it instead of just adding related references.

Response: We thank the reviewer for drawing our attention to this point. Stress relaxation is the observed decrease in stress in response to strain generated in the structure. This is primarily due to keeping the structure in a strained condition for some finite interval of time and hence causing some amount of plastic strain. Because synthetic hydrogel materials are viscoelastic, they have the property of stress relaxation to relieve stress under constant strain. We believed this was the reason why PAM single-network and PAM/PEG double-network hydrogels weakened apparently after pre-stretching or pre-compressing.

5. The SEM image of “stretched” liposomes is much less convincing than before. What is shown in Figure S13 seems like some wrinkles of polymers or air pockets, not the liposomes. The authors also claimed that the size of liposomes increased after stretching. How is it possible? Shouldn't the volume of the liposomes remain the same and only the morphologies change? And how many SEM images were used for the size distribution in Figure S13? It looks like it was obtained based on one SEM image which is not convincing. And it appears that the large-sized liposomes the authors counted are just larger wrinkles.

Response: We thank the reviewer for highlighting this issue. SEM test was reperfomed and a clearer image was substituted in Figure S13a. Similar to images reported in the previous publications (e.g. Carbohyd. Polym. 2015, 115, 651 and Biomacromolecules 2014, 15, 3587), the vesicles observed in the SEM image were liposomes other than wrinkles of polymers or air pockets. We agree with the reviewer's point that the volume of the liposomes could remain the same, while the morphologies changed after stretching. The average diameter of the embedded liposomes in the same direction that the gel was stretched was obtained through analyzing about 420 liposomes in several SEM images (Figure S13b).

6. When the authors showed a picture of a 100g weight hanging on the hydrogel, does it match their tensile strength measurement? What was the dimension of the hydrogel that was used for the experiment? Was the hydrogel still fully hydrated?

Response: Figure 5b, II was just a digital photo of an SSH strip (25 × 4 × 1.5 mm) bearing a

100 g weight, which was used to demonstrate that a solid gel on the basis of the second network was formed after swelling. The tensile strength of the second network was shown in Figure 5c. The hydrogel was fully hydrated throughout the experiment.

Reviewer #2

Article by Wu et al. reported that mechanical strength of hydrogels can be enhanced at higher swelling ratio because reactive precursors in vesicle can be released by swelling-induced stretching and form new polymer network. It is interesting to incorporate vesicles containing clickable monomers into hydrogel network for enhancing their mechanical property when it was stretched upon swelling without further treatment.

However, as authors mentioned already in the main text, Gong et al. (ref. 27) reported swelling- or stretching-induced double network formation inside hydrogel, in which swelling-strengthen phenomena can be even further continued upon external stimuli just like muscle. By contrast, the hydrogels in this report was strengthen upon swelling once but not any more than once.

Therefore, I believe that authors need to justify why this single-use swelling-strengthening hydrogels should be developed or find proper application. Swelling-strengthening phenomenon itself would not be justification for publication in this journal because there are many examples of stronger hydrogels in swollen state including Polymers 2018, 10, 1025 (doi:10.3390/polym10091025).

But, again, this work may be well received in specialized journal such as biomaterials or Advanced healthcare materials. It would be interesting to encapsulate tetrafunctional monomers in separate vesicle.

Response: We thank the reviewer for taking his/her time to review our manuscript. Also, we are grateful that the reviewer acknowledges that our work is interesting. It is necessary to point out that ref. 27 by Gong et al (Science 2019, 363, 504–508) actually reports stretching-induced strengthening, which is different from our work that describes swelling-strengthening hydrogels. Meanwhile, the mentioned reference (Polymers 2018, 10, 1025) reports a tough nanocomposite hydrogel with swelling-resistant and anti-dehydration properties in water-glycerol bi-solvent solutions, which is also irrelevant to the concept of swelling-strengthening. As we stated in the previous point-by-point response, the necessity of multiple-strengthening capability really depended on its specific applications. When hydrogels are applied in physiological environments, such as biological glues or artificial tissues, a given mechanical strength is required. In these situations, it is critical to avoid the loss of mechanical strength after swelling. By virtue of the characteristic of swelling-strengthening, our system shows potential to address these challenges.

Reviewer #4

The authors have adequately addressed the concerns raised by Reviewer 3 in the first review. The method presented is interesting and solves the issue that hydrogels weaken with swelling. There are applications for this work in biological applications, which are supported by the biocompatibility results presented in the article.

Response: We thank the reviewer very much for his/her positive review of our work.

I will reiterate that the clarity of the manuscript should be improved. For example:

- In the first sentence, "being" should be "are".

Response: It has been corrected in the revised manuscript.

- Line 32, "hyperaemia" should be defined or omitted. For example, the sentence could be written: "For example, muscles become much stronger after swelling with blood (6,7)."

Response: This has been modified accordingly in the revised version.

- It is not clear how hyperaemia swelling leads to increased stiffness. Instead, the mechanism of hyperaemia swelling itself is emphasized (e.g., Lines 32-37). This is akin to describing the swelling process of synthetic gels without mentioning that the swelling process weakens the gel by diluting the network. Why does muscle tissue not swell-weaken like synthetic hydrogels? The polymer network (e.g., collagen fibers) similarly dilutes upon swelling. Understanding this mechanism is important to the method presented in this paper.

Response: We thank the reviewer for bringing this point to our attention. We will try our best to address this insightful comment. Actually, the mechanism of hyperaemia-induced strengthening of muscle is quite complicate. A recent paper has delivered some important hints (Nature, 2019, DOI: 10.1038/s41586-019-1516-5). While, the existence of many biological membrane barriers that can regulate the selective transmembrane transport of water molecules and ions is critical for swelling-induced strengthening of muscle. It means that during hyperaemia swelling muscle can maintain a steady structure of the network. Differently, the network of typical synthetic hydrogels can be diluted upon swelling and hence decrease the mechanical strength. Inspired by the biological membrane barriers, liposomal membrane barriers that covalently embedded in a crosslinked network have been used to regulate the transmembrane transport. After swelling, new network can be formed and therefore the mechanical strength of the initial gel can be maintained.

- Line 36, "increasement" should be "increase".

Response: It has been corrected in the revised manuscript.

- Line 54, "initiate" should be "initiates".

Response: We have addressed this accordingly.

- Line 152, "highly efficient via a catalyst-free reaction" could instead be "via a highly efficient catalyst-free reaction".

Response: This has been modified accordingly.

- Line 204, "deigned" should be "designed".

Response: This typo has been corrected in the revised manuscript.

We thank all the reviewers again for taking their valuable time to review our manuscript. These useful comments and constructive suggestions are highly appreciated.

Reviewers' Comments:

Reviewer #1:

Remarks to the Author:

After a careful review, this manuscript still requires further revisions. Below is the detailed critique.

1. English. There are still NUMEROUS grammar errors throughout the manuscript. Below are just a few examples.

Line 45, "Studies have been proven..." -> "Studies have shown.."

Line 48, "The key challenge is that, during swelling the major change occurred is..." -> "the key challenge is that during swelling, the major change that occurs is..."

Line 67, " The double bonds decorated liposomes were covalently incorporated.." -> "The liposomes decorated with double bonds were covalently..."

Line 90, "Even with the swelling ratio increased up to 75%, the SSH could remain its initial mechanical strength" -> "Even with a 75% increase in swelling ratio, the SSH could reTAIN its initial mechanical strength".

Line 91, "However, a typical swelling weakening phenomenon was observed to the control hydrogels." -> "However, a typical swelling weakening phenomenon was observed IN the control hydrogels."

Line 95, "Further, the compressive stress of the SSH at 90% strain raised from 1.32 ± 0.04 to 1.71 ± 0.04 MPa with the swelling ratio increased to 25%" -> "Further, the compressive stress of the SSH at 90% strain INCREASED from 1.32 to 1.71 MPa AS the swelling ratio increased BY 25%"

2. Regarding the decreased stiffness after compression/stretching. The authors wrote, "while the corresponding PAM single-network and PAM/PEG double-network hydrogels weakened apparently due to their viscoelasticity that could cause stress relaxation". It does not seem that the authors understand the mechanism accurately. Stretching or compressing does not cause "stress relaxation" in the covalently crosslinked hydrogels. It only adds stress to the polymer chains. Stress relaxation comes into play after the stretch or compression is removed. In reference 40, the term "stress relaxation" was used to explain the hysteresis during the repeated strain-stress measurements, but it did not mean the actual weakening of the material. In fact, the point of the paper was more about release of water during the compression and insufficient time for re-swelling. It is highly recommended that the authors re-think the mechanism.

On a related note, what happens to the macroscopic shape of the hydrogel after stretching? The authors mentioned the stretching was performed for several minutes. If the covalent bonds are formed in a stretched state, wouldn't that alter the final shape of the hydrogel?

3. Volume of the liposomes after stretching. The authors wrote the following in their response, "We agree with the reviewer's point that the volume of the liposomes could remain the same, while the morphologies changed after stretching." However, the authors still maintained the same stance in the manuscript, stating that the stretching increased the volume of the liposomes (which was manifested by an increase in the average diameter). I still have a hard time understanding why one would expect an increase in volume of liposomes after stretching. The authors provided new SEM images of liposomes (Fig S13) after stretching. They should also provide such high-resolution images of liposomes before stretching in to confirm their hypothesis.

Reviewer #4:

Remarks to the Author:

The authors have adequately addressed the comments raised by Reviewer 3 (as mentioned in the previous review). The clarity of the manuscript has been improved according to the cases I raised in the previous round, but, from what I can tell, has not been improved otherwise.

REVIEWER COMMENTS

Reviewer #1 (Remarks to the Author):

1. English. There are still NUMEROUS grammar errors throughout the manuscript. Below are just a few examples.

Line 45, "Studies have been proven..." -> "Studies have shown.."

Line 48, "The key challenge is that, during swelling the major change occurred is..." -> "the key challenge is that during swelling, the major change that occurs is..."

Line 67, " The double bonds decorated liposomes were covalently incorporated.." -> "The liposomes decorated with double bonds were covalently..."

Line 90, "Even with the swelling ratio increased up to 75%, the SSH could remain its initial mechanical strength" -> "Even with a 75% increase in swelling ratio, the SSH could reTAIN its initial mechanical strength".

Line 91, "However, a typical swelling weakening phenomenon was observed to the control hydrogels." -> "However, a typical swelling weakening phenomenon was observed IN the control hydrogels."

Line 95, "Further, the compressive stress of the SSH at 90% strain raised from 1.32 ± 0.04 to 1.71 ± 0.04 MPa with the swelling ratio increased to 25%" -> "Further, the compressive stress of the SSH at 90% strain INCREASED from 1.32 to 1.71 MPa AS the swelling ratio increased BY 25%"

Response: According to the reviewer's suggestions, these have been modified in our revised manuscript. The authors believe that the majority of these are different fashions of writing.

2. Regarding the decreased stiffness after compression/stretching. The authors wrote, "while the corresponding PAM single-network and PAM/PEG double-network hydrogels weakened apparently due to their viscoelasticity that could cause stress relaxation". It does not seem that the authors understand the mechanism accurately. Stretching or compressing does not cause "stress relaxation" in the covalently crosslinked hydrogels. It only adds stress to the polymer chains. Stress relaxation comes into play after the stretch or compression is removed. In reference 40, the term "stress relaxation" was used to explain the hysteresis during the repeated strain-stress measurements, but it did not mean the actual weakening of the material. In fact, the point of the paper was more about release of water during the compression and insufficient time for re-swelling. It is highly recommended that the authors re-think the mechanism.

On a related note, what happens to the macroscopic shape of the hydrogel after stretching? The authors mentioned the stretching was performed for several minutes. If the covalent bonds are formed in a stretched state, wouldn't that alter the final shape of the hydrogel?

Response: We are grateful to the reviewer for her/his insightful comments. The authors would like to highlight that stress relaxation is just a potential explanation for the weakening

phenomenon of the control experiments of PAM single-network and PAM/PEG double-network hydrogels after pre-stretching or pre-compressing. Variation of the network in hydrogels could be much more complicated during pre-stretching or pre-compressing. Viscoelastic gels under constant strain exhibit partial stress relaxation, or a decrease in the stress required to maintain the strain over time (Nature Materials 2016, 15, 326; Biomaterials 2018, 154, 213; Acta Biomaterialia 2017, 62, 82). The relevant references have been added as ref. 40-42 in the revised version. In addition, as the second network formed in the stretched state was relatively weaker than the first network, which was insufficient to alter the final macroscopic shape of the hydrogel. This is in good agreement with our recent work published in Advanced Materials 2020, 32, 1906870.

3. Volume of the liposomes after stretching. The authors wrote the following in their response, "We agree with the reviewer's point that the volume of the liposomes could remain the same, while the morphologies changed after stretching." However, the authors still maintained the same stance in the manuscript, stating that the stretching increased the volume of the liposomes (which was manifested by an increase in the average diameter). I still have a hard time understanding why one would expect an increase in volume of liposomes after stretching. The authors provided new SEM images of liposomes (Fig S13) after stretching. They should also provide such high-resolution images of liposomes before stretching in to confirm their hypothesis.

Response: Actually, we described that the average size of the embedded liposomes increased after SWELLING other than after STRETCHING. The stretched polymer network during swelling loosened the lipid bilayer arrangement of the embedded liposomes, which led to an increase in the volume of the liposomes. According to the reviewer's request, a high-resolution image of liposomes without stretching and swelling has been supplemented in Figure 4e.

Reviewer #4 (Remarks to the Author):

The authors have adequately addressed the comments raised by Reviewer 3 (as mentioned in the previous review). The clarity of the manuscript has been improved according to the cases I raised in the previous round, but, from what I can tell, has not been improved otherwise.

Response: We appreciate the reviewer's comments.

Reviewers' Comments:

Reviewer #1:

Remarks to the Author:

1. For the comment on English grammar and spelling, the authors responded, "The authors believe that the majority of these are different fashions of writing." These are NOT stylistic preferences but are clear grammar/spelling errors. It is really surprising that the authors have not cleared up these errors when they submit their manuscript to a renowned international journal. Below are just a couple of many examples. I did not bother to pick all the errors because that is not my responsibility.

Line 52: "Liposomal membrane nanobarriers that covalently embedded ..." -> "Liposomal membrane nanobarriers that ARE covalently embedded.."

Line 97: "With the swelling ratio further increased up to 75%, the compressive stress was able to maintain at the same level of the gel before swelling" -> "With the swelling ratio further increasing up to 75%, the compressive stress remained at the same level of the gel before swelling"

2. The authors still failed to explain the decrease in modulus of the control hydrogels after pre-stretching. Although this is not the main theme of this manuscript, the fact that the authors cannot explain this phenomenon scientifically and rather rely on vague words (e.g. viscoelasticity, stress relaxation), which the authors do not seem to have good understanding of. What is the mechanism of stress relaxation for the covalently crosslinked hydrogels? The new references the authors added in the revision are basically from the same group, and do not support the experimental results of this manuscript. The cited papers mention that covalently crosslinked hydrogels do not relax applied stresses effectively, while the presence of physical crosslinks provides the mechanisms for stress relaxation. It is very concerning that the authors keep adding references that do not really support their results.

REVIEWER COMMENTS

Reviewer #1 (Remarks to the Author):

1. For the comment on English grammar and spelling, the authors responded, "The authors believe that the majority of these are different fashions of writing." These are NOT stylistic preferences but are clear grammar/spelling errors. It is really surprising that the authors have not cleared up these errors when they submit their manuscript to a renowned international journal. Below are just a couple of many examples. I did not bother to pick all the errors because that is not my responsibility.

Line 52: "Liposomal membrane nanobarriers that covalently embedded ..." -> "Liposomal membrane nanobarriers that ARE covalently embedded."

Line 97: "With the swelling ratio further increased up to 75%, the compressive stress was able to maintain at the same level of the gel before swelling" -> "With the swelling ratio further increasing up to 75%, the compressive stress remained at the same level of the gel before swelling"

Response: On the basis of the reviewer's suggestion, the language of the manuscript has been polished by a native English-speaking colleague.

2. The authors still failed to explain the decrease in modulus of the control hydrogels after pre-stretching. Although this is not the main theme of this manuscript, the fact that the authors cannot explain this phenomenon scientifically and rather rely on vague words (e.g. viscoelasticity, stress relaxation), which the authors do not seem to have good understanding of. What is the mechanism of stress relaxation for the covalently crosslinked hydrogels? The new references the authors added in the revision are basically from the same group, and do not support the experimental results of this manuscript. The cited papers mention that covalently crosslinked hydrogels do not relax applied stresses effectively, while the presence of physical crosslinks provides the mechanisms for stress relaxation. It is very concerning that the authors keep adding references that do not really support their results.

Response: To avoid potential disputation and ambiguity regarding the underlying mechanism of the control hydrogels after pre-stretching, we have only focused on the experimental phenomenon and results in the revised manuscript.